# Prevalence of Arterial Stiffness Determined by Cardio-Ankle Vascular Index in Myeloproliferative Neoplasms [note 1]

**DOI:** 10.3390/jcm14196944

**Published:** 2025-09-30

**Authors:** Thanakharn Jindaluang, Ekarat Rattarittamrong, Chatree Chai-Adisaksopha, Pokpong Piriyakhuntorn, Lalita Norasetthada, Adisak Tantiworawit, Thanawat Rattanathammethee, Sasinee Hantrakool, Nonthakorn Hantrakun, Teerachat Punnachet, Piangrawee Niprapan, Siriluck Gunaparn, Arintaya Phrommintikul

**Affiliations:** 1Department of Internal Medicine, Nakornping Hospital, Chiang Mai 50180, Thailand; thanakharn.jin@cpird.in.th; 2Division of Hematology, Department of Internal Medicine, Faculty of Medicine, Chiang Mai University, Chiang Mai 50200, Thailand; chatree.chai@cmu.ac.th (C.C.-A.); pokpong.p@cmu.ac.th (P.P.); lalita.n@cmu.ac.th (L.N.); adisak.tan@cmu.ac.th (A.T.); thanawat.r@cmu.ac.th (T.R.); sasinee.h@cmu.ac.th (S.H.); nonthakorn.h@cmu.ac.th (N.H.); teerachat.pun@cmu.ac.th (T.P.); piangrawee.n@cmu.ac.th (P.N.); siriluck.g@cmu.ac.th (S.G.); arintaya.p@cmu.ac.th (A.P.)

**Keywords:** arterial stiffness, cardio-ankle vascular index, myeloproliferative neoplasms, polycythemia vera, essential thrombocythemia, primary myelofibrosis

## Abstract

**Objective**: This study investigated the prevalence of arterial stiffness among individuals diagnosed with myeloproliferative neoplasms (MPNs), specifically essential thrombocythemia (ET), polycythemia vera (PV), and primary myelofibrosis (PMF). **Methods**: We performed a cross-sectional study at Chiang-Mai University Hospital, Thailand, defining arterial stiffness as a mean cardio-ankle vascular index (CAVI) ≥8.0. Patients were compared to age-, sex-, and Thai cardiovascular (CV) risk score-matched controls with CV risk factors. Additional outcomes included the 10-year CV risk in MPN patients, estimated by the Thai CV risk score, and the correlation between plasma C-reactive protein (CRP) levels and CAVI. **Results**: Eighty participants were included (50 with PV, 24 with ET, 6 with PMF; median age: 63.5 years). Arterial stiffness was present in 63.8% of MPN patients overall, with respective rates for ET, PV, and PMF being 70.8%, 60.0%, and 66.7% (*p* = 0.655). When compared to matched non-MPN controls with CV risk, prevalence of arterial stiffness did not differ significantly (65.2% vs. 60.9%, *p* = 0.539). The median estimated 10-year CV risk for patients with MPNs was 13.6% (range 0.7–30.0). No significant association was observed between CRP levels and mean CAVI (R = 0.208, *p* = 0.073). **Conclusions**: Arterial stiffness was detected in 63.8% of individuals with MPNs, a prevalence like that of matched non-MPN patients with CV risk factors.

## 1. Introduction

Myeloproliferative neoplasms (MPNs), particularly essential thrombocythemia (ET), polycythemia vera (PV), and primary myelofibrosis (PMF), are clonal disorders that result from driver genetic abnormalities, including *JAK2*V617F, *CALR*, or *MPL* mutation, leading to unregulated proliferation and differentiation of myeloid progenitor cells [1]. About 30–40% of MPN patients have thrombotic events which are the most serious complication of MPNs [2]. Many clinical and laboratory features, including age older than 60 years, history of thrombosis, high white blood cell count, and *JAK2*V617F mutation, were described as risk factors for thrombosis [3,4,5,6,7].

Arterial stiffness is an indicator of atherosclerosis [8,9] and can be used for the early detection of premature atherosclerosis [10]. Cardio-ankle vascular index (CAVI) is a diagnostic tool to evaluate overall arterial stiffness from the aortic origin to the ankle, independent of blood pressure. The accuracy and sensitivity of CAVI to evaluate arterial stiffness have been widely performed [10,11,12,13]. Previous studies and systematic reviews indicated the correlation between high CAVI values and cardiovascular outcomes [10,11,12]. The optimal cut-off CAVI value of ≥8.0 to 9.0 is associated with coronary artery disease and cardiovascular disease [13,14,15]. CAVI was evaluated in addition to the Thai cardiovascular (CV) risk score to predict ten-year CV risk in the Thai population. This study showed a high prevalence of coronary artery disease of more than 60% in a very high-risk group (CAVI ≥ 8.0), compared to less than 10% in patients with CAVI < 8.0 [14].

The link between hematologic malignancies and arterial stiffness was observed mainly in patients receiving chemotherapy, tyrosine kinase inhibitors, radiotherapy, or hematopoietic stem cell transplantation [16]. In addition, parameters in complete blood count (CBC) such as anemia, leukocytosis, and high mean platelet volume were also reported as risk factors of arterial stiffness in hematologic malignancies [16]. However, there was limited data regarding arterial stiffness, ten-year cardiovascular risk determined by Thai CV risk score, and correlation between inflammatory markers and CAVI in MPN patients. Therefore, this study aimed to determine the prevalence of arterial stiffness determined by CAVI, defined ten-year CV risk as well as the correlation between CAVI and plasma C-reactive protein (CRP) in patients with MPNs, including ET, PV, and PMF.

## 2. Materials and Methods

### 2.1. Study Design and Patients

This study was a cross-sectional study conducted at Maharaj Nakorn Chiang Mai Hospital, Chiang Mai University, Chiang Mai, Thailand between 11th August 2020 to 10th August 2021. Eligible patients were more than 18 years of age with a diagnosis of ET, PV, or PMF according to World Health Organization (WHO) 2016 diagnostic criteria [1]. Since *MPL* mutation testing was not available in our institute, the diagnosis of ET in case of unmutated *JAK2*V617F and *CALR* was based on the first 3 major criteria and 1 minor criterion. Likewise, we used the presence of another clonal marker or the absence of reactive myelofibrosis as the third major criterion for the diagnosis of PMF without *JAK2*V617F and *CALR* mutation [1]. Patients with the following characteristics were not eligible: severe heart failure or aortic insufficiency, atrial fibrillation, peripheral arterial disease in which ankle-brachial index (ABI) <0.9, and patients who received alpha1-adrenergic receptor blocker. All ET, PV, or PMF patients without these exclusion criteria were eligible regardless of treatment and duration of disease.

### 2.2. Outcomes

The primary outcome was the prevalence of arterial stiffness evaluated by CAVI in patients with MPNs, including ET, PV, and PMF. The secondary outcomes were ten-year CV risk in MPN patients calculated by Thai CV risk score as well as the correlation between plasma CRP and CAVI among patients with MPNs.

### 2.3. Procedures

Written informed consent was obtained from eligible patients. Subsequently, participants were scheduled for CAVI measurement and plasma CRP testing. Demographic data, including age, sex, cardiovascular risk factors, history of thrombosis, and co-morbid diseases were collected and calculated for the Thai CV risk score. Information about MPN diagnosis, laboratory data, risk group, and treatment were also gathered. The definition of “high risk MPNs” used in this study was age ≥60 years or having a history of thrombosis. The subgroup analysis according to risk group was performed for the primary outcome.

CAVI was measured using the VaSera CAVI instrument MA-300HDS(V) by Fukuda Denshi Company, LTD (Tokyo, Japan), software version 06-04. Patients were placed in the supine position, and pulse wave velocity (PWV) between the heart and ankles was measured. Electrocardiogram and cardiac phonographic were monitored. The calculation was performed using the following formula [10,11,15]:CAVI = (2ρ/ΔP) × [ln(P_s_/P_d_) ] × PWV^2^
where P_s_: systolic pressure, P_d_: diastolic pressure, PWV: pulse wave velocity (PWV = L/T, where L is the distance between the aortic valve to the ankle, and T is time for the pulse wave to propagate from the aortic valve to the ankle), ΔP: pulse pressure (P_s_ − P_d_), and ρ: blood density of 1.5 g/mL. CAVI values were performed bilaterally, and mean CAVI was calculated, using cutoff values of mean CAVI ≥8.0 to determine arterial stiffness. This cut-off was recommended by the Physiological Diagnosis Criteria for Vascular Failure Committee [16]. CAVI was performed by an experienced research nurse who was blinded to the participants’ clinical data.

The prevalence of arterial stiffness in MPNs was compared to previous data on age-, sex- and Thai CV risk score-, and propensity score-matched Thai non-MPN patients with CV risk factors at a 1:3 ratio. The data of Thai non-MPN patients with CV risk derived from the cohort of patients with high risk for cardiovascular events (CORE) registry was a prospective, multicenter, observational, longitudinal study of Thai patients aged 45 years or older with established atherosclerotic disease, which is defined as coronary artery disease, cerebrovascular disease, peripheral arterial disease, or had at least three atherosclerosis risk factors [17].

Ten-year CV risk had been assessed by Thai CV risk score using the online calculator available on https://www.rama.mahidol.ac.th/cardio_vascular_risk/thai_cv_risk_score/ (accessed on 10 July 2021). The model variables consist of age, sex, history of smoking, diabetes, hypertension, and total cholesterol level (mg/dL). If total cholesterol was not available, waist circumference (inches) was considered. The risk was reported as a percentage of 10-year CV risks. Plasma CRP was analyzed by the technique of nephelometry.

### 2.4. Statistical Analysis

We assumed that MPN patients may have a prevalence of arterial stiffness of about 60% [14]. The study would require a sample size of 77 for estimating the expected proportion, with 11% absolute precision and 95% confidence. The prevalence of arterial stiffness was determined by CAVI, and 10-year cardiovascular risk calculated by Thai CV risk score was analyzed by descriptive analysis. Comparisons of continuous variables were analyzed by independent *t*-test, and categorical variables were analyzed by the Chi-square test. The correlation between plasma CRP and CAVI values was analyzed using Pearson correlation. All analyzed data were performed using STATA version 14 (Stata Corp., College Station, TX, USA). A *p*-value of less than 0.05 was considered statistically significant.

## 3. Results

### 3.1. Patients’ Characteristics

Overall, 82 MPN patients were screened, and two patients were excluded due to ABI < 0.9, resulting in 80 patients being enrolled and evaluated for CAVI. These 80 enrolled patients included 24 ET (30.0%), 50 PV (62.5%), and six PMF patients (7.5%). The clinical characteristics of the patients are shown in Table 1. The median age was 63.5 years (range 29–90), and 37 patients (46.3%) were male. Sixty-eight patients (85.0%) had *JAK2*V617F mutation, including 15 patients (62.5%) with ET, 49 patients (98%) with PV, and four patients (66.7%) with PMF. Five patients (6.3%) had *CALR* mutation, and seven patients (8.8%) had non-mutated *JAK2* V617F and *CALR*. There were 61 of 80 patients (76.3%) who were high risk MPNs. Most patients received treatment with aspirin [67 patients (83.8%)] as antithrombotic therapy and received hydroxyurea as a cytoreductive therapy [65 patients (81.3%)], whereas only four patients (5.0%) received anagrelide. There were 17 patients (21.3%) who had history of thrombosis, including stroke [11 patients (13.8%)], myocardial infarction [five patients (6.3%)], peripheral arterial disease [one patient (1.3%)], and portal vein thrombosis [one patient (1.3%)].

### 3.2. Outcomes

The prevalence of arterial stiffness determined by CAVI ≥8 in patients with MPNs was 63.8% (95% confidential interval [CI] 52.2–74.2) (51 from 80 patients). There was no significant difference in the prevalence of arterial stiffness among ET (70.8%), PV (60.0%), and PMF (66.7%) in which mean ± SD CAVI of 8.5 ± 1.3, 8.3 ± 1.2, and 8.7 ± 1.7, respectively (*p* = 0.459) (Table 2).

The prevalence of arterial stiffness in MPN patients was compared to age-, sex-, and Thai CV risk score-matched control non-MPN patients with cardiovascular risk, consisting of 66 patients for the case group and 197 patients for the control group (Table 3). Clinical characteristics of the match-control population include median age of 63.4 years, male 53.8%, hypertension 95.9%, diabetes 58.8%, smoking 4.1%, and median ten-year Thai CV risk score was 14.7%. Prevalence of arterial stiffness in MPN patients compared to control was 65.2% (95% CI 52.4–76.4) and 60.9% (95% CI 53.7–67.7) (*p* = 0.539), in which mean CAVI were 8.5 ± 1.3 and 8.4 ± 1.4 (*p* = 0.487), respectively.

High-risk MPN patients had a higher mean CAVI than low-risk MPNs (8.6 ± 1.2 and 7.6 ± 1.0, respectively, *p* = 0.002). As a result, high-risk MPN patients had more prevalence of arterial stiffness compared to low-risk MPN patients, 70% (95% CI 51.4–81.4) and 42% (95% CI 20.2–66.5), respectively (*p* = 0.025) (Figure 1).

Analysis of mean CAVI in subgroups of patients is shown in Table 4. Patients with age ≥60 years had a higher mean CAVI than patients with age <60 years (8.8 ± 0.2 and 7.6 ± 0.2, respectively) (*p* < 0.005). There was no statistically significant difference in mean CAVI between patients who had and did not have a history of thrombosis (7.9 ± 0.9 and 8.5 ± 1.3, respectively) (*p* = 0.055), nor in mutation status, time from diagnosis, WBC and platelet count, and the response of treatment. Prevalence of arterial stiffness among patients who received aspirin was 67.2% (95% CI 54.6–78.1), compared to patients who did not receive aspirin 46.2% (95% CI 19.2–74.8) (*p* = 0.149) in which means CAVI of 8.5 ± 1.2 and 7.9 ± 1.4, respectively (*p* = 0.103). There was also a non-statistic difference in the prevalence of arterial stiffness between patients who received and did not receive hydroxyurea, 66.2% (95% CI 53.3–77.4) vs. 53.3% (95% CI 26.5–78.7) (*p* = 0.352).

Multivariable regression from dataset before propensity score matching (80 MPN and 408 non-MPN patients) with controlling for arterial stiffness risk factors as confounders including hypertension, diabetes, and smoking showed MPN was not an independent risk factor to arterial stiffness (CAVI ≥8.0) (odds ratio 1.48 [0.67–3.28], *p* = 0.328).

The median ten-year CV risk calculated by Thai CV risk score in MPN patients was 13.6% (range 0.7–30.0). Median (range) of 10-year CV risk among patients with ET, PV, and PMF was 10.9% (0.7–30.0), 15.1% (0.9–30.0), and 11.0% (8.7–23.2), respectively (*p* = 0.844). Comparison of 10-year CV risk (median, range) between patients with and without a history of thrombosis showed 15.2% (0.7–28.9) and 13.3% (0.7–30.0), respectively (*p* = 0.492). The patients with high risk MPNs had significantly higher 10-year CV risk (median, range) than low-risk MPNs [16.0% (0.7–30) vs. 3.4% (0.7–24.8), respectively, *p* < 0.001].

Plasma CRP had been measured in 75 of 80 patients (93.8%). Median CRP was 3.1 mg/L (range 0.6–21.3). Correlation between CRP and mean CAVI was not significant (R = 0.208, *p* = 0.073). Median (range) CRP level in patients who had arterial stiffness by using CAVI ≥ 8.0 was 3.1 mg/L (1.1–21.3), compared with 2.9 mg/L (0.6–10.2) in cases with CAVI < 8.0 (*p* = 0.145). Median (range) CRP level between patients with and without a history of thrombosis was 2.9 mg/L (0.8–17.7) and 3.1 mg/L (0.6–21.3), respectively (*p* = 0.674). Comparison of CRP level (median, range) between patients with high-risk and low-risk MPNs showed no statistically significant difference [3.1% (0.8–21.3) and 3.1% (0.6–8.9), respectively, *p* = 0.465].

## 4. Discussion

This study demonstrated that the prevalence of arterial stiffness in patients with MPNs determined by the mean CAVI ≥8.0 was 63.8%. There was no statistical difference in the prevalence of arterial stiffness between ET, PV, and PMF. However, this subgroup analysis should be interpreted with caution. The mean CAVI cut-off values of ≥8.0 were associated with arterial stiffness, cardiovascular disease, and significant excess mortality and had been validated in the Thai population [14,20]. The CAVI values were higher among patients who had CV risk factors (e.g., hypertension, diabetes, dyslipidemia, smoking) [10,11,12]. A systematic review from a total of 26 studies, mostly in Asian countries, indicated that higher CAVI values were associated with CV disease [12]. The present study also showed that patients with MPNs had a prevalence of arterial stiffness as high as non-MPN patients with CV risk. This result inferred that MPNs are possibly the risk factor that can contribute to a CV disease resembling high atherosclerotic risk patients. However, multivariable regression from the dataset before matching did not demonstrate MPN as an independent risk factor to arterial stiffness.

MPN patients who have had more than one CV risk factor had a significantly poor survival outcome [21]. A large prospective cohort of PV patients reported that age ≥65 years and a history of thrombosis were associated with CV events and mortality [5]. The study in ET patients also revealed that age ≥60 years, previous vascular complications, and hypercholesterolemia were associated with major vascular events [6]. Regarding PMF, age older than 60 years and presence of *JAK2* V617F mutation were significant risk factors of thrombosis, while the presence of CV risk factor tended to increase risk (hazard ratio 2.25, 95%CI 0.90–5.61, *p* = 0.08) [22]. As a result, this study used age 60 ≥ years or history of thrombosis for determining high risk MPNs. This study showed patients with high-risk MPNs had significantly higher mean CAVI and prevalence of arterial stiffness than low-risk MPNs, consequently higher probability of CV events. This finding was likely due to older age rather than a history of thrombosis, since the subgroup analysis showed that mean CAVI in patients with age ≥60 years was significantly higher than younger patients, whereas patients who had previous thrombosis tended to have lower mean CAVI. As a result, high-risk MPN patients, especially age ≥60 years, should be closely monitored and investigated for other CV risk factors together with lifestyle modification.

A recent systematic review and meta-analysis revealed the pooled prevalence of thrombosis in MPN patients at diagnosis was 20.0%. The most frequent CV events in MPN patients were ischemic stroke (7.4%), coronary artery disease (6.1%), transient ischemic attack (3.5%), and deep vein thrombosis (3.4%). PV patients had more prevalence of thrombosis (28.6%) than ET (20.7%) and PMF (9.5%) [23]. However, there was scarcely any data on 10-year CV risk in MPN patients that were reported by Framingham risk score or Thai CV risk score. Frederiksen H et al. demonstrated that MPN patients had five-year vascular disease from 0.5 to 7.7% and increased the rate of vascular disease by 1.3 to 3.7 times compared with the matched general population [24]. This study showed that the ten-year CV risk calculated by the Thai CV risk score in MPN patients was 13.6%. Patients with PV tended to have a higher 10-year CV risk compared to those with ET and PMF (15.1%, 10.9%, and 11.0%, respectively). The ten-year CV risk in MPN patients with a history of thrombosis was also slightly higher than in those without a history of thrombosis (15.2% and 13.3%, respectively). Long-term follow-up of these patients is required for validation of Thai CV risk score in MPN patients.

Chronic inflammation is important pathophysiology in arterial stiffness. The association between arterial stiffness and elevation of CRP level has been discovered [25,26,27]. Previous studies demonstrated that CRP levels are also elevated in patients with MPNs [3,28]. This study could not show the correlation between mean CAVI and CRP levels in patients with MPNs. This might be explained by many reasons. Firstly, patients who were included in the present study had a median time from diagnosis of more than five years and had variabilities in response to treatment that might affect the CRP level. Secondly, plasma CRP, which is not sensitive as high sensitivity CRP (hs-CRP), was used to measure inflammation in this study due to lack of availability. As a result, the authors cannot exclude the role of circulating inflammation on arterial stiffness of MPN patients. Future studies to analyze the correlation between hs-CRP or novel inflammatory biomarkers and CAVI should be performed.

The strength of this study is the prospective study and using CAVI to determine arterial stiffness in patients with MPNs. Previously, only limited data were available according to this aspect. The CAVI was selected to measure arterial stiffness in this study because of its promising role in terms of the prediction of coronary artery disease in Thai population [14]. Changes in CBC parameters were reported to be associated with increase CAVI in hematologic malignancies [16]. As a result, CAVI was an interesting investigation to determine arterial stiffness in MPN. Further studies regarding therapeutic intervention to decrease CAVI such as anti-hypertensive, lipid-lowering, and anti-diabetic medications in MPN patients are warranted as in high-risk non-MPN individuals [29].

On the contrary, there were several limitations. Firstly, mean CAVI values in this study were measured by cross-sectional evaluation that had limitations for causal inference. Selection bias could have occurred due to some exclusion criteria such as ABI < 0.9. Variability data of time from diagnosis, treatment of MPNs, the response of treatment as well as socioeconomic and inflammatory status might have affected CAVI values and arterial stiffness. Improvement and controlling of CV risk factors from lifestyle modification (e.g., smoking cessation, exercise) or medication (e.g., renin-angiotensin-aldosterone system inhibitors, calcium channel blockers, statins) can lower CAVI values [12]. Thus, the effect of treatment in patients with MPNs and the change in CAVI values should be explored in a future prospective study. Secondly, CV risk calculated by the Thai CV risk score was the estimated ten-year CV risk, prospective data with long-term follow-up to observe cardiovascular events should be considered. Finally, there was no comparison of the prevalence of arterial stiffness with a normal population without CV risk since there was no available cohort in Thai population. A prospective study to compare mean CAVI between MPN patients and the general population might be needed to answer this question.

## 5. Conclusions

The prevalence of arterial stiffness determined by CAVI in patients with MPNs was 63.8% and was comparable with a non-MPN population with CV risk. Patients with high-risk MPNs defined by age ≥60 years or history of thrombosis had higher mean CAVI and may exhibit more arterial stiffness than low-risk MPNs. The association between arterial stiffness and the effect of treatment in patients with MPNs should be explored by future prospective study.

## Figures and Tables

**Figure 1 jcm-14-06944-f001:**
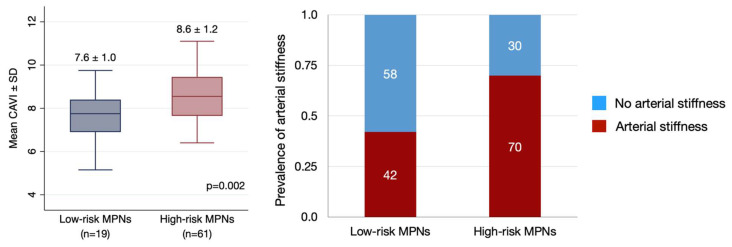
Comparison of mean cardio-ankle vascular index (CAVI) (**left**) and prevalence of arterial stiffness determined by CAVI ≥8 (**right**) between high-risk myeloproliferative neoplasms (MPNs) and low-risk MPNs. (High-risk MPNs defined by age ≥60 years or history of thrombosis).

**Table 1 jcm-14-06944-t001:** Clinical characteristics of patients.

Characteristics	Total (*n* = 80)	ET (*n* = 24)	PV (*n* = 50)	PMF (*n* = 6)
Age (mean ± SD), years	63.5 ±12.6	60.9 ±16.2	64.0 ± 11.0	70.3 ± 5.1
Sex, n (%)				
Male	37 (46.3)	9 (37.5)	26 (52.0)	2 (33.3)
Female	43 (53.8)	15 (62.5)	24 (48.0)	4 (66.7)
Mutation, n (%)				
*JAK2* V617F	68 (85.0)	15 (62.5)	49 (98.0)	4 (66.7)
*CALR*	5 (6.3)	4 (16.7)	0	1 (16.7)
Non-*JAK2* V617F and *CALR*	7 (8.8)	5 (20.8)	1 (2.0)	1 (16.7)
History of smoking, *n* (%)	20 (25.0)	7 (29.2)	13 (26.0)	0
Comorbidity, n (%)				
Hypertension	33 (41.3)	7 (29.2)	23 (46.0)	3 (50.0)
Diabetes	11 (13.8)	7 (29.2)	4 (8.0)	0
Treatment, n (%)				
Hydroxyurea	65 (81.3)	18 (75.0)	45 (90.0)	2 (33.3)
Aspirin	67 (83.8)	20 (83.3)	47 (94.0)	0
Anticoagulant	3 (3.8)	1 (4.2)	2 (4.0)	0
Anagrelide	4 (5.0)	2 (8.3)	2 (4.0)	0
Ruxolitinib	1 (1.3)	0	0	1 (16.7)
Clopidogrel	3 (3.8)	1 (4.2)	2 (4.0)	0
History of thrombosis ^+^, n (%)	17 (21.3)	5 (20.8)	12 (24.0)	1 (16.7)
Venous DVT or PE	0	0	0	0
Portal vein thrombosis	1 (1.3)	0	1 (2.0)	0
Arterial MI	5 (6.3)	2 (8.3)	3 (6.0)	0
Stroke	11 (13.8)	2 (8.3)	8 (16.0)	1 (16.7)
PAD	1 (1.3)	1 (4.2)	0	0
Time from diagnosis, median (range), years	5.5 (0.1–17.4)	4.0 (0.2–13.6)	6.0 (0.1–17.4)	3.0 (0.4–6.3)
Waist circumference * (mean ± SD), inches	31.9 ± 4.4	31.9 ± 5.5	31.9 ± 4.0	30.8 ± 3.7
C-reactive protein level ^#^, median (range), mg/L	3.1 (0.6–21.3)	3.0 (1.0–12.4)	3.1 (0.6–17.7)	3.0 (2.9–21.3)
Total Cholesterol (mean ± SD), mg/dL	152.8 ± 35.6	155.0 ± 37.3	154.4 ± 32.9	130.7 ± 49.4
LDL (mean ± SD), mg/dL	85.0 ± 29.2	83.1 ± 34.4	87.7 ± 24.2	69.7 ± 43.4
HDL (mean ± SD), mg/dL	44.8 ± 13.2	49.1 ± 15.4	44.0 ± 10.7	34.2 ± 16.4
CBC at date of diagnosis (mean ± SD)				
Hemoglobin, g/dL	15.9 ± 3.6	12.6 ± 2.0	18.1 ± 2.1	10.3 ± 2.3
Hematocrit, %	49.7 ± 6.9	38.9 ± 5.7	57.0 ± 6.9	32.7 ± 6.8
WBC count, × 10^9^/L	17.0 ± 8.8	12.5 ± 4.1	17.6 ± 8.4	29.9 ± 12.7
Platelet count, × 10^9^/L	836.1 ± 433.1	119.2 ± 406.1	723.7 ± 340.2	349.3 ± 227.1
CBC at date of enrollment (mean ± SD)				
Hemoglobin, g/dL	12.6 ± 2.3	12.4 ± 1.8	13.2 ± 2.1	8.7 ± 1.7
Hematocrit, %	38.3 ± 6.4	37.5 ± 5.3	40.0 ± 5.9	28.1 ± 5.0
WBC count, × 10^9^/L, median (range)	7.9 (3.0–62.0)	6.0 (3.0–30.1)	9.5 (3.4–53.6)	18.7 (5.6–62.0)
Platelet count, × 10^9^/L	488.1 ± 271.8	589.4 ± 310.1	473.0 ± 239.0	208.3 ± 141.0

^+^ One patient had a history of two previous thromboses. * Missing data for waist circumference for one patient. ^#^ Seventy-five of 80 patients (93.8%) had been analyzed for CRP level. ET: essential thrombocythemia; PV: polycythemia vera; PMF; primary myelofibrosis; SD: standard deviation; DVT: deep vein thrombosis; PE: pulmonary embolism; MI: myocardial infarction; PAD: peripheral arterial disease; LDL: low-density lipoprotein; HDL: high-density lipoprotein; CBC: complete blood count.

**Table 2 jcm-14-06944-t002:** Prevalence of arterial stiffness determined by cardio-ankle vascular index ≥ 8 and 10-years cardiovascular (CV) risk calculated by Thai CV risk score in patients with myeloproliferative neoplasms.

Results	Total (*n* = 80)	ET (*n* = 24)	PV (*n* = 50)	PMF (*n* = 6)	*p* Value
Prevalence of arterial stiffness, *n* (%)	51 (63.8)	17 (70.8)	30 (60.0)	4 (66.7)	0.655
CAVI (mean ± SD)	8.4 ± 8.3	8.5 ± 1.3	8.3 ± 1.2	8.7 ± 1.7	0.459
10-yr CV risk by Thai CV risk score, %, median (range)	13.6 (0.7–30.0)	10.9 (0.7–30.0)	15.1 (0.9–30.0)	11.00 (8.7–23.2)	0.844

ET: essential thrombocythemia; PV: polycythemia vera; PMF; primary myelofibrosis; SD: standard deviation; IQR: interquartile range; CAVI: cardio-ankle vascular index; CV: cardiovascular.

**Table 3 jcm-14-06944-t003:** Clinical characteristics of propensity score–matched patients.

	MPNs Patients (*n* = 66)	Matched Control (*n* = 197)	*p* Value
Age, median (IQR), years	64.3 (59–71)	63.4 (56–71)	0.574
Male, *n* (%)	34 (51.5)	106 (53.8)	0.747
10-year CV risk by Thai CV risk score, median (IQR)	16.0 (9.6–23.9)	14.7 (8.2–21.0)	0.960
CAVI, mean ± SD	8.5 ± 1.3	8.4 ± 1.4	0.487
Presence of arterial stiffness, *n* (%)	43 (65.2)	120 (60.9)	0.539

MPN: myeloproliferative neoplasm; IQR: interquartile range; CV: cardiovascular; CAVI: cardio-ankle vascular index; SD: standard deviation.

**Table 4 jcm-14-06944-t004:** Results of mean cardio-ankle vascular index in subgroup of patients.

Results	N (%)	CAVI (Mean ± SD)	*p* Value
Myeloproliferative neoplasms			0.459
ET	24 (30.0)	8.5 ± 1.3	
PV	50 (62.5)	8.3 ± 1.2	
PMF	6 (7.5)	8.7 ± 1.7	
Mutation status			0.474
*JAK2* V617F	68 (81.3)	8.3 ± 1.3	
*CALR*	5 (6.3)	8.4 ± 0.9	
Non-*JAK2* or *CALR*	7 (8.8)	8.6 ± 1.6	
Time from diagnosis			0.253
Less than 1 year	12 (15.0)	8.2 ± 1.5	
2–5 years	27 (33.7)	8.5 ± 1.4	
6–10 years	28 (35.0)	8.1 ± 0.9	
More than 11 years	13 (16.3)	8.8 ± 1.3	
Age			<0.005
≥60 years	53 (66.3)	8.8 ± 0.2	
<60 years	27 (33.7)	7.6 ± 0.2	
History of thrombosis			0.055
Thrombosis	17 (21.3)	7.9 ± 0.9	
No thrombosis	58 (78.7)	8.5 ± 1.3	
WBC counts at diagnosis			0.978
<10 × 10^9^/L	12 (15.0)	8.4 ± 1.1	
≥10 × 10^9^/L	68 (85.0)	8.3 ± 1.3	
WBC counts at CAVI performed			0.121
<10 × 10^9^/L	51 (63.8)	8.2 ± 1.2	
≥10 × 10^9^/L	29 (36.3)	8.7 ± 1.3	
Platelet counts at diagnosis			0.844
<450 × 10^9^/L	17 (21.3)	8.4 ± 1.4	
≥450 × 10^9^/L	63 (78.8)	8.4 ± 1.2	
Platelet counts at CAVI performed			0.844
<450 × 10^9^/L	44 (55.0)	8.4 ± 1.4	
≥450 × 10^9^/L	36 (45.0)	8.4 ± 1.2	
Response of treatment ET *			0.965
Complete remission	7 (29.2)	8.4 ± 1.7	
Partial remission	6 (25.0)	8.6 ± 1.2	
No response	11 (45.8)	8.5 ± 1.2	
Response of treatment PV *			0.344
Complete remission	19 (38.0)	8.1 ± 1.5	
Partial remission	14 (28.0)	8.1 ± 1.2	
No response	17 (34.0)	8.6 ± 1.2	
Response of treatment PMF *			0.672
Clinical improvement	1 (16.7)	7.1	
Stable disease	2 (33.3)	9.3 ± 1.5	
Symptom response	3 (50.0)	8.8 ± 2.1	
Treatment with aspirin			0.103
Yes	67 (83.7)	8.5 ± 1.2	
No	13 (16.3)	7.9 ± 1.4	
Treatment with hydroxyurea			0.579
Yes	65 (81.2)	8.3 ± 1.2	
No	15 (18.3)	8.5 ± 1.5	

* Criteria for assessing response to treatment according to European LeukemiaNet criteria for PV and ET [18] and the International Working Group for Myelofibrosis Research and Treatment (IWG-MRT) criteria for PMF [19]. CAVI: cardio-ankle vascular index; SD: standard deviation; ET: essential thrombocythemia; PV: polycythemia vera; PMF; primary myelofibrosis; WBC: white blood cells

## Data Availability

The data that supports the findings of this study are available from the corresponding author, [E.R.], upon reasonable request.

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
