# Peer review of "Prevalence of Arterial Stiffness Determined by Cardio-Ankle Vascular Index in Myeloproliferative Neoplasms†"

_jcm, 2025, doi:10.3390/jcm14196944_

Round 1

Reviewer 1 Report

Comments and Suggestions for Authors

Dear Authors,

I want to congratulate you on conducting this excellent study.

This manuscript investigates arterial stiffness in patients with MPNs using the CAVI, comparing prevalence against non-MPN controls. The study addresses an important knowledge gap regarding cardiovascular complications in patients with MPN. However, there are several methodological and analytical issues that need some clarification:

  • The type of study, cross-sectional, cannot be used to confirm the causality between MPNs and arterial stiffness. Therefore, the authors need to justify more in the discussion why this relationship exists.
  • The authors recruited non-MPN as controls, however, these participants have several cardiovascular risk factors, which may refute their hypothesis as MPN is an independent risk factor to an increased arterial stiffness. How would you justify this?
  • Please provide more detail on inclusion/exclusion criteria, especially on the treatment history of hydroxyurea, ruxolitinib.
  • It is unclear from the methods who performed CAVI measurements, and whether they blind to participant’ clinical data?
  • The authors measured only CRP in this study, which is not sensitive as hs-CRP. Therefore, the authors cannot calim that increased circulating inflammation is associated with increased arterial stiffness. Although, the authors mention that as a limitation, they need to re-write this part in the discussion. Given that, there is no correlation between CRP and CAVI, the authors need to explain why? and expand in this part.
  • I would recommend the authors to re-analysis their data and run multivariable regression with controlling for arterial stiffness risk factors as confounders (i.e. hypertension, diabetes, smoking, and treatment effects).
  • The authors used CAVI ≥ 8.0 as a cut-off value. Is there a justification for this?
  • In the results, you need to add confidence intervals to prevalence comparisons, and be cautious with subgroup analyses, as the study is not powered to conduct subgroup analyses.

Wish you all the best,

Author Response

Response to reviewer 1

This manuscript investigates arterial stiffness in patients with MPNs using the CAVI, comparing prevalence against non-MPN controls. The study addresses an important knowledge gap regarding cardiovascular complications in patients with MPN. However, there are several methodological and analytical issues that need some clarification:

Comment 1:

  • The type of study, cross-sectional, cannot be used to confirm the causality between MPNs and arterial stiffness. Therefore, the authors need to justify more in the discussion why this relationship exists.

Response 1: This point is added to the Discussion part as limitation of study. “Firstly, mean CAVI values in this study were measured by cross-sectional evaluation that had limitations for causal inference. Variability data of time from diagnosis, treatment of MPNs, and the response of treatment might be affecting the CAVI values and arterial stiffness.”

Comment 2:

  • The authors recruited non-MPN as controls, however, these participants have several cardiovascular risk factors, which may refute their hypothesis as MPN is an independent risk factor to an increased arterial stiffness. How would you justify this?

Response 2: We agree with your comment. So, in the Discussion part, the statement was change to “The present study also showed that patients with MPNs had a prevalence of arterial stiffness as high as non-MPNs patients with CV risk. This result inferred that MPNs are possibly the risk factor that can contribute to a CV disease resembling high atherosclerotic risk patients. However, multivariable regression from dataset before matching did not demonstrate MPN as an independent risk factor to arterial stiffness.”  

Comment 3:

  • Please provide more detail on inclusion/exclusion criteria, especially on the treatment history of hydroxyurea, ruxolitinib.

Response 3: More detail regarding this point was added in Methods part - Study Design and Patients. “All ET, PV, or PMF patients without these exclusion criteria were eligible regardless of treatment and duration of disease.”

Comment 4:

  • It is unclear from the methods who performed CAVI measurements, and whether they blind to participant’ clinical data?

Response 4: The information regarding this point was added to the Methods part - Procedure. “CAVI was performed by an experienced research nurse that blinded to participants’ clinical data.”

Comment 5:

  • The authors measured only CRP in this study, which is not sensitive as hs-CRP. Therefore, the authors cannot claim that increased circulating inflammation is associated with increased arterial stiffness. Although, the authors mention that as a limitation, they need to re-write this part in the discussion. Given that, there is no correlation between CRP and CAVI, the authors need to explain why? and expand in this part.

Response 5: More discussion regarding inflammatory markers and arterial stiffness was addressed. “This study could not show the correlation between mean CAVI and CRP levels in patients with MPNs. These might be explained by many reasons. Firstly, patients who were included in the present study had a median time from diagnosis of more than five years and had variabilities in response to treatment that might affect the CRP level. Secondly, plasma CRP which is not sensitive as high sensitivity CRP (hs-CRP) was used to measure inflammation in this study. As a result, the authors cannot exclude the role of circulating inflammation on arterial stiffness of MPN patients. Future studies to analyze the correlation between hs-CRP or novel inflammatory biomarkers and CAVI should be performed.” 

Comment 6:

  • I would recommend the authors to re-analysis their data and run multivariable regression with controlling for arterial stiffness risk factors as confounders (i.e. hypertension, diabetes, smoking, and treatment effects).

Response 6: Multivariable regression was analyzed and included in Result part. “Multivariable regression from dataset before propensity score matching (80 MPN and 408 non-MPN patients) with controlling for arterial stiffness risk factors as confounders including hypertension, diabetes, and smoking showed MPN was not an independent risk factor to arterial stiffness (CAVI > 8.0) (odds ratio 1.48 [0.67-3.28], p=0.328).”

Comment 7:

  • The authors used CAVI ≥ 8.0 as a cut-off value. Is there a justification for this?

Response 7: This point was mentioned in Method part – Procedure.  “This cut-off was recommended by the Physiological Diagnosis Criteria for Vascular Failure Committee [16].”

Comment 8:

  • In the results, you need to add confidence intervals to prevalence comparisons, and be cautious with subgroup analyses, as the study is not powered to conduct subgroup analyses.

Response 8: The confidential intervals were added in Result part. “Prevalence of arterial stiffness in MPNs patients compared to control was 65.2% (95% CI 52.4-76.4) and 60.9% (95% CI 53.7-67.7) (p = 0.539)” The caution was mentioned in Discussion part. “There was no statistical difference in the prevalence of arterial stiffness between ET, PV, and PMF. However, this subgroup analysis should be interpreted with caution.”

Reviewer 2 Report

Comments and Suggestions for Authors

The manuscript addresses an important and clinically relevant topic: the prevalence of arterial stiffness, as measured by the CAVI in patients with MPNs. The study is timely and contributes to the growing body of literature on cardiovascular risk in hematological malignancies. The prospective design, use of a validated vascular assessment tool and inclusion of a matched control group are commendable strengths.

Comments

  1. The introduction provides a reasonable background but could be strengthened by incorporating more recent and diverse literature on arterial stiffness in hematological disorders beyond MPNs.

  2. The rationale for focusing on CAVI, as opposed to other arterial stiffness indices, should be more explicitly justified in the context of MPN pathophysiology.

  3. While the cross‑sectional design is acknowledged, the discussion should more explicitly address its limitations for causal inference, particularly regarding the temporal relationship between MPN diagnosis, treatment exposure, and arterial stiffness development.

  4. The exclusion of patients with ABI < 0.9 is reasonable, but the potential for selection bias should be discussed.

  5. Please clarify the matching methodology (exact matching vs. propensity score) and provide evidence of baseline comparability between groups beyond age, sex, and Thai CV risk score.

  6. The results are clearly presented, but some p‑values are reported without effect sizes or confidence intervals; these should be included for interpretability.

  7. The discussion should more deeply explore the clinical implications of finding similar arterial stiffness prevalence in MPN patients and high‑risk non‑MPN individuals.

  8. The absence of association between CRP and CAVI warrants a more nuanced discussion, including possible explanations and consideration of other inflammatory biomarkers.

  9. The English is overall understandable but could be refined for clarity, concision, and grammatical precision.

  10. Consider discussing residual confounding from unmeasured variables such as medication use, inflammatory status, or socioeconomic factors.

Author Response

Response to reviewer 2

The manuscript addresses an important and clinically relevant topic: the prevalence of arterial stiffness, as measured by the CAVI in patients with MPNs. The study is timely and contributes to the growing body of literature on cardiovascular risk in hematological malignancies. The prospective design, use of a validated vascular assessment tool and inclusion of a matched control group are commendable strengths.

Comments

Comment 1: The introduction provides a reasonable background but could be strengthened by incorporating more recent and diverse literature on arterial stiffness in hematological disorders beyond MPNs.

Response 1: The background of arterial stiffness in hematologic malignancies was added in the introduction. “The link between hematologic malignancies and arterial stiffness was observed mainly in patients receiving chemotherapy, tyrosine kinase inhibitors, radiotherapy, or hematopoietic stem cell transplantation [16]. In addition, parameters in complete blood count such as anemia, leukocytosis, and high mean platelet volume were also reported as risk factors of arterial stiffness in hematologic malignancies [16].”

Comment 2: The rationale for focusing on CAVI, as opposed to other arterial stiffness indices, should be more explicitly justified in the context of MPN pathophysiology.

Response 2: The rationale in MPN patients was added in the Discussion part. “The CAVI was selected to measure arterial stiffness in this study because the promising role of CAVI in terms of prediction of coronary artery disease in Thai population [14]. Since changes in CBC parameters were reported to be associated with increase CAVI in hematologic malignancies [16]. As a result, CAVI was an interesting investigation to determine arterial stiffness in MPN.”

Comment 3: While the cross‑sectional design is acknowledged, the discussion should more explicitly address its limitations for causal inference, particularly regarding the temporal relationship between MPN diagnosis, treatment exposure, and arterial stiffness development.

Response 3: This point is added to the Discussion part as limitation of study. “Firstly, mean CAVI values in this study were measured by cross-sectional evaluation that had limitations for causal inference. Selection bias could be occurred from some exclusion criteria such as ABI < 0.9. Variability data of time from diagnosis, treatment of MPNs, the response of treatment as well as socioeconomic and inflammatory status might be affecting the CAVI values and arterial stiffness.”  

Comment 4: The exclusion of patients with ABI < 0.9 is reasonable, but the potential for selection bias should be discussed.

Response 4: This point is also added to the Discussion part as limitation of study as the same paragraph of Response 3.

Comment 5: Please clarify the matching methodology (exact matching vs. propensity score) and provide evidence of baseline comparability between groups beyond age, sex, and Thai CV risk score.

Response 5: The matching methodology as added in Methods part - Procedures “The prevalence of arterial stiffness in MPNs was compared to previous data on age-, sex- and Thai CV risk and propensity score matched Thai non-MPN patients with CV risk with a 1:3 ratio.” Table 3 was added to demonstrate the comparability between groups.

Comment 6: The results are clearly presented, but some p‑values are reported without effect sizes or confidence intervals; these should be included for interpretability.

Response 6: This issue was added in revised manuscript.

Comment 7: The discussion should more deeply explore the clinical implications of finding similar arterial stiffness prevalence in MPN patients and high‑risk non‑MPN individuals.

Response 7: The discussion regarding this aspect was added. “Further studies regarding therapeutic intervention to decrease CAVI such as anti-hypertensive, lipid-lowering, and anti-diabetic medications in MPN patients are warranted as in high‑risk non‑MPN individuals [29].” 

Comment 8: The absence of association between CRP and CAVI warrants a more nuanced discussion, including possible explanations and consideration of other inflammatory biomarkers.

Response 8: More discussion regarding inflammatory markers and arterial stiffness was addressed. “This study could not show the correlation between mean CAVI and CRP levels in patients with MPNs. These might be explained by many reasons. Firstly, patients who were included in the present study had a median time from diagnosis of more than five years and had variabilities in response to treatment that might affect the CRP level. Secondly, plasma CRP which is not sensitive as high sensitivity CRP (hs-CRP) was used to measure inflammation in this study. As a result, the authors cannot exclude the role of circulating inflammation on arterial stiffness of MPN patients. Future studies to analyze the correlation between hs-CRP or novel inflammatory biomarkers and CAVI should be performed.” 

Comment 9: The English is overall understandable but could be refined for clarity, concision, and grammatical precision.

Response 9: The authors revised the English grammar and concision in this manuscript.

Comment 10: Consider discussing residual confounding from unmeasured variables such as medication use, inflammatory status, or socioeconomic factors.

Response 10: This point is added to the Discussion part as limitation of study. “On the contrary, there were several limitations. Firstly, mean CAVI values in this study were measured by cross-sectional evaluation that had limitations for causal inference. Selection bias could be occurred from some exclusion criteria such as ABI < 0.9. Variability data of time from diagnosis, treatment of MPNs, the response of treatment as well as socioeconomic and inflammatory status might be affecting the CAVI values and arterial stiffness.”

Round 2

Reviewer 1 Report

Comments and Suggestions for Authors

No further comments.